# How Has the COVID-19 Pandemic Affected the Way We Access and Interact with the Countryside and the Animals within It?

**DOI:** 10.3390/ani11082281

**Published:** 2021-08-02

**Authors:** Jo Hockenhull, Keith Squibb, Amelia Cameron

**Affiliations:** 1Animal Welfare and Behaviour Group, Bristol Veterinary School, University of Bristol, Langford, Bristol BS40 5DU, UK; amelia.cameron@nottingham.ac.uk; 2Independent Researcher, Bristol BS40 5DU, UK; keith.squibb@hotmail.com; 3School of Veterinary Medicine and Science, University of Nottingham, Loughborough LE12 5RD, UK

**Keywords:** animal welfare, feeding, human behaviour, nature, wildlife

## Abstract

**Simple Summary:**

In March 2020, the UK entered its first mandatory lockdown to reduce the spread of COVID-19. The restrictions associated with the lockdown led to changes in human behaviour, particularly in relation to exercise and accessing the outdoors. This study used an online survey of UK residents to explore these changes and to understand how people interacted with the natural world while they were outside, including interactions with wildlife and domestic animals. There was an increase in how often respondents went for walks outside during lockdown, compared to pre-lockdown levels, and this increase lasted beyond the end of the first lockdown. Interacting with animals was a common feature of walks. This most commonly involved watching wildlife or domestic animals, such as livestock or horses, but sometimes also involved physically interacting with them and/or feeding them which may have implications for their welfare, for example, if inappropriate food is provided. There is also a risk of humans transferring disease between the animals they have contact with or zoonotic disease transmission between the humans and animals. While is it positive that people are interacting with the natural world, it is important that this does not compromise animal wellbeing.

**Abstract:**

There is growing evidence that the changes in human behaviour resulting from the COVID-19 pandemic have had positive and negative impacts on the natural world. This study used an online survey to explore how the first UK lockdown affected human exercising behaviour, with particular focus on the role of wild and domestic animals. The survey was completed by 308 respondents. There was a significant increase in the frequency that respondents went for walks outdoors during lockdown, in comparison to pre-lockdown levels (*p* ≤ 0.001), and this was sustained (albeit to a lesser extent) once lockdown ended (*p* = 0.005). Engaging with the natural world was an important feature of walks outside for 81% of respondents. A small proportion of respondents reported physically interacting with the animals they encountered and/or feeding them, which may have implications for their welfare. The findings suggest that those who value animal encounters during their time outside always seek these interactions, while those who do not, did not tend to change this behaviour following lockdown. Should the changes in human exercising behaviour be sustained, it is important to balance the benefits of walking outdoors for human health and wellbeing with the health and welfare of the animals they encounter.

## 1. Introduction

The COVID-19 pandemic led to dramatic changes in human behaviour across the globe, as national governments brought in mandatory lockdowns of some or all of their population in an attempt to control the spread of the virus [1]. This rapid reduction in human activity (and the associated noise, traffic, and pollution) brought with it the opportunity for global wildlife to fill the voids left behind and exploit the environs that the presence of humans and/or their associated pollution had previously rendered inaccessible [2,3,4]. Anecdotal reports abounded in the national and international media of nature reclaiming towns and cities (for example, see [5,6]. Whether or not they were true (the claim that dolphins had been sighted in the canals of Venice was later found to be falsified [7]), they gave people across the globe positive news and pleasure at a time of great uncertainty.

Lockdown led to other changes in human behaviour, particularly in relation to exercising. During the first lockdown in the UK (23 March–10 May 2020), people were only allowed outside of their homes to exercise once a day, with government officials recommending this lasted up to one hour. This opportunity to spend time outside of the house became incredibly valuable to many. The benefits of physical exercise, in combination with time outdoors, on mental health wellbeing are gaining increasing recognition [8,9,10,11]. While it is likely that some people engaged in more outdoor exercising behaviour during lockdown to benefit their wellbeing, others may have stayed inside due to fear of viral spread, should they come into contact with people from other households.

The mass change in human activity patterns is likely to have impacted the natural environment. Anecdotal reports of increasing litter, as well as interactions between walkers and domestic animals (resulting in some animals being inadvertently harmed by being fed by passers-by), suggest that not all of these impacts were positive for the animals involved. Indeed, some sources expressed concerns for animal populations within urban ecosystems that typically depend on human beings for their food, such as city dwelling pigeons, rats, and sea gulls [2].

The impact of the change in human behaviour, as a result of COVID-19, on the natural world is complex and little understood at present, although the body of evidence suggesting both positive and negative effects is growing [2,3,12]. This study contributes to the growing evidence base, using an online survey to explore human activity during the initial lockdown in the UK. The study had three main objectives: to quantify how lockdown affected human exercising behaviour (walking) and use of green spaces, to investigate what people did on their walks (including their interactions with wild and domestic animals (livestock and horses)), and to better understand what features of the walks were important for the people taking them during lockdown.

## 2. Methods

### 2.1. Ethical Approval

This project received approval from the University of Bristol Faculty of Health Research Ethics Committee (Ref: 104762).

### 2.2. Survey Design

An online survey consisting of 20 questions (16 closed (tick one option only or all options that apply, depending on the question) and four open text boxes for respondents to give more detailed answers. The survey was accessed from a homepage containing participant information and the criteria for participation (i.e., the participant must be a UK resident aged 18 years or over). The respondents were required to tick an agreement to participate before being allowed to proceed with the survey. This was the only forced response question in the whole survey, the remaining questions were optional. A definition of countryside was provided on the homepage as follows: “In the context of this questionnaire countryside refers to any general green areas such as fields, parks, woodlands, forests etc. but not your private back garden”. The full survey can be accessed in the Appendix A and is briefly summarised below.

The first page of the survey consisted of four closed-questions requesting background information on the respondent, including their age, the area of the UK in which they live, and the type of area they live in (e.g., rural or city). A tick-all-that-apply question requested details on the respondent’s access to outside spaces (e.g., balconies, allotments, and gardens).

The second page of the survey contained seven closed-questions exploring respondents’ use of green spaces before lockdown, during lockdown, and after the restrictions were eased (key timepoints in the UK, relating to COVID-19 restrictions, are summarized in Figure 1). This section also asked questions relating to whether respondents ever strayed from marked footpaths and whether they had noticed any changes in the amount of litter they encountered on walks.

The third page focused on any interactions with wildlife, livestock, and horses while out walking. These three groups of animals were chosen based on anecdotal reports of humans engaging with these types of animals whilst on walks. In the UK, horses are not considered a livestock species, so they were grouped separately in the survey. Three tick-all-that-apply questions explored the types of interactions, including none, observing, physically interacting, and feeding; three open questions requested further details on these interactions and whether the interactions had changed since lockdown.

The final page of the survey contained two tick-all-that-apply questions to capture what respondents enjoy about walking and what health benefits it may have. The final question was open-ended for any additional relevant comments the respondents may have.

The survey was pilot tested (n = 4) prior to going live. No changes to format or wording were required following this process. It took the pilot test participants 4–5 min to complete the survey, and this information was provided on the survey homepage to inform potential respondents of the anticipated time commitment.

### 2.3. Subject Recruitment

The survey details were shared via social media (Facebook), by email, and by posting in the University of Bristol Yammer social chat, as well as the Bristol Veterinary School newsletter. The survey was also shared on the survey sharing website SurveyCircle (https://www.surveycircle.com/en/ accessed on 23 September 2020). The survey was live between 23 September and 4 November 2020.

### 2.4. Analysis

Data were exported in a CSV file from the Jisc Online Survey platform and opened in Microsoft Excel for Microsoft 365, where incomplete entries were removed (those that only ticked the initial question to agree to the terms of participation), alongside those where the data were questionable, due to the answers provided in the open text boxes. Closed-question answer options were coded into numerical values and exported to IBM SPSS Statistics (version 26) (IBM, New York, USA). As there were relatively few free-text responses (only 1.3–24.4% of the respondents used the open text boxes, depending on the question) and respondents kept their answers very short, these data were viewed in Excel to explore common answers that added details to the closed-question responses, rather than undergoing thematic analysis.

The quantitative data underwent descriptive analysis in SPSS, using the frequencies function to ascertain the distribution of responses. As the survey was comprised of closed, categorical questions, the data were not normally distributed; this was confirmed by the visual inspection of the histograms and Q-Q plots alongside the Kolmogorov-Smirnov and Shapiro-Wilk tests, both of which indicated that the data did not meet the criteria for normality (*p* ≤ 0.001). Consequently, non-parametric tests were chosen for analysis. Changes in walking behaviour pre-, during, and post-lockdown were analysed statistically using a Friedman Test, and post-hoc Dunn tests with a Bonferroni correction were run between each pairing to explore where any differences in walking frequency were between these three time-points. Chi-square tests, for independence, were used to investigate whether interactions with wildlife, livestock, horses, and other animals were associated with walking alone, walking with other adults, or walking with children.

## 3. Results

The survey was completed by 311 respondents. Three response sets were removed (one incomplete and two due to questions of data accuracy), leaving data from 308 respondents. The majority of respondents were from England (n = 278, 91.1%) and 3.0% (n = 9) were from Wales, 5.2% (n = 16) were from Scotland, and 0.7% (n = 2) were from Northern Ireland. The median age of respondents was 35–44 years. All types of locations were represented within the sample, with 23.1% (n = 71) of the respondents living in the city, 4.9% (n = 15) living in the inner city, 44.5% (n = 137) living in a suburb or town, 20.8% (n=64) living in a village, and 6.8% (n = 21) living in a rural location.

Most respondents (92.2%, n = 284) had access to some type of outside space, with many reporting that they had one or more of the options provided in the question (Figure 2).

The frequency that respondents went for walks differed significantly between reported levels pre-lockdown, during lockdown, and post-lockdown (χ^2^_2_ = 59.358, *p* ≤ 0.001; Table 1). Dunn post-hoc tests, with a Bonferroni correction, showed that walking frequency increased significantly during (*p* ≤ 0.001) and post-lockdown (*p* = 0.005), in comparison to pre-lockdown levels.

Respondent walking behaviour is presented in Table 2. Most respondents reported that since the UK entered lockdown, they have walked closer to home. The majority of respondents (64.3%, n = 198) walked with another adult and/or on their own (55.2%, n = 170). The majority of respondents did not walk with children (83.1%, n = 256). Only 26.3% (n = 81) of respondents reported that they never stray off the main path.

The interactions that respondents reported having with animals on their walks is presented in Table 3. The majority of the respondents (85.1%, n = 262) reported watching wildlife on their walks, while 66.9% (n = 206) watched other animals (including livestock and horses), with 30.2% (n = 93) reporting that they watched animals more since lockdown. While 10.1% (n = 31) of the respondents reported feeding wildlife on walks, 4.2% (n = 13) reported feeding horses, 0.6% (n = 2) fed livestock, and 1.6% (n = 5) fed other animals (for example, dogs, swans, and geese).

Who respondents walked with did not significantly affect their interactions with wildlife, livestock, or other animals. However, respondents who walked alone (χ^2^_1_ = 4.387, *p* = 0.036) or with children (χ^2^_1_ = 6.252, *p* = 0.012) were more likely to feed horses than those who walked with other adults.

Few respondents reported negative impacts of walking on their physical (1%, n = 3) or mental (1%, n = 3) health, although 18.8% (n = 58) reported no change and 5.5% (n = 17) were unsure whether their walks had benefitted them or not (Table 4). Respondents enjoyed their time outside for a variety of reasons (see Table 4). Those who described ‘other’ reasons to those listed in the question mentioned factors, such as the space away from other people, sense of freedom, time alone, escapism, foraging for edibles, the opportunity for photography, and immersion in the sights, smells, and sounds of nature as reasons they enjoyed their time outside.

## 4. Discussion

This study provides an insight into the outdoor walking behaviour of a sample of UK residents during the first mandatory COVID-19 lockdown (March–May 2020) and the role that animal encounters played in this activity. Respondents spanned the full range of geographic locations (in terms of country within the UK and the type of area they lived in), although the sample was skewed towards respondents living in suburbs or towns in England. This skew in the sample likely stems from the recruitment strategy used, which was biased towards English social media groups and mailing lists, as these were more easily accessed by the research team, who were all based in England. It was positive to note that most respondents had access to some kind of outdoor space, with the majority having a private garden. We recognise, however, that the skewed nature of the sample may have influenced our finding. Data from the Office of National Statistics (ONS) suggests that 12% of British households had no access to a private or shared garden in 2020 (during the pandemic); this rose to 21% for households in cities [13].

During lockdown, Natural England’s *The People and Nature Survey for England* reported that walking was by far the most popular outdoor activity, with over 70% of their sample undertaking this activity [14]. The second most popular activity was wildlife-watching, undertaken by just under a quarter of those sampled [14]. In the present study, respondents similarly reported increasing the frequency they went for walks outdoors during lockdown, and while this decreased in the post-lockdown period, it remained significantly higher than pre-lockdown levels. The survey was conducted 4–6 months following the end of the first lockdown, and it was positive to see that many respondents reported sustaining their outdoor walking this far beyond lockdown. It is important to note, however, that during lockdown, there was also an increase in the percentage of respondents reporting that they never go for walks outdoors, although the values remained low (from 0.7% pre-lockdown to 4.9% during lockdown). Like wearing a mask, hand washing, and social distancing, limiting time spent outside was a health-protective/risk-mitigating behaviour used by some individuals to reduce their risk of contracting COVID-19 [15,16]. However, as suggested by another survey-based study, it appears that some of the respondents used the opportunity provided by the COVID-19 restrictions to increase their daily physical activity from their pre-COVID levels [17].

Where respondents walked was also reported to have changed since lockdown, with more people walking closer to home and in more rural or remote areas; this was most likely a result of the government lockdown restrictions on travel. The change to walking in more rural and remote areas is worth noting, as it is possible that these respondents may not have been familiar with The Countryside Code and, consequently, their behaviour may have had a greater impact on the animals and environment they encountered. Thirty-one percent of respondents reported that they had explored a wider range of locations (when walking and spending time outside) than they did previously. Most respondents walked alone or with one other adult.

Nearly 75% of respondents reported that they sometimes strayed off the main paths they were following, most commonly to gain access to areas they would otherwise be unable to reach. Only a small percentage (9%) said that this was to enable them to interact with wildlife, horses, or livestock. Straying off marked paths can appear harmless but may have significant repercussions, for example, if it results in trespassing (which is currently a civil offence), disrupts habitats, or disturbs ground-nesting birds (e.g., [18]).

Interactions with animals appeared to be a common component of respondent’s walks, with only 3.9% reporting that they do not interact with wildlife in any way: 85% observed wildlife, while 10% reported feeding wild animals. In comparison, more respondents said that they do not interact with domestic animals on their walks, although 66.9% reported observing them. Only a small percentage of respondents reported feeding domesticated animals: 4.2% reported feeding horses, 0.6% fed livestock, and 1.6% fed other domestic animals, the most common examples provided being ducks and swans. Only feeding horses was significantly associated with who respondents walked with. Those who walked alone or with children were more likely to feed horses than those who walked with other adults. This finding is comparable to that of a similar study exploring public interactions with privately owned horses, where the majority of respondents reported that most of the people they saw feeding their horses without permission were families [19].

The welfare implications of feeding and physically interacting with wild and domestic animals can be significant. Food is often used to lure animals closer so they can be observed more easily or so that tactile contact can be made. For example, one respondent reported using food to bait camera traps for badgers. Details were not provided as to what physical interactions with wildlife entailed. It would be beneficial to explore this further, provided the harm that may inadvertently be done by handling wild animals, especially the young (which would have been abundant during the initial lockdown which spanned spring in the UK). Often the food that is offered to animals by humans is not appropriate and this may have consequences for health and survival. At the time the survey was conducted, The Countryside Code did not explicitly state not to feed horses and livestock. However, it has since been updated with the statement “Do not feed livestock, horses or wild animals as it can cause them harm”, as well as requesting that wild animals, livestock, and horses be provided plenty of space [20]. The Countryside Code was also updated in the summer of 2020, as a response to issues raised during lockdown (in that instance, increases in littering and dogs worrying sheep [21]), demonstrating the impact that COVID-related changes in human exercising behaviour was having on the countryside.

The public received mixed messages regarding feeding animals they encounter. For example, feeding wild birds is often actively encouraged, particularly during winter, providing the feedstuff is appropriate [22]. Although concerns have been raised that such supplemental feeding results in dependency and increases the spread of infectious diseases, the evidence that this is the case is often not clear cut [23,24,25,26], and typically bird feeding has been associated with improved health and survival rates [25,27]. Feeding other species of wild or free-ranging animals is generally not as common as feeding garden birds, although there are indicators that this is changing, e.g., with the increased availability of food for hedgehogs [28]. Public information campaigns have worked hard in recent years to discourage the feeding of bread to waterfowl, as well as feeding bread and milk to hedgehogs, due to the risks involved from ingesting this inappropriate foodstuff. However, appropriate alternatives, such as green vegetables and wheat grain for waterfowl [29] and non-fish-based cat food for hedgehogs [28] are often recommended instead. Both of these campaigns, and the growing market of food for garden wildlife, strongly suggests that it is *what* is fed, rather than the action of feeding, that is the problem.

In contrast, feeding privately owned domestic animals, for example, neighbourhood cats [30], while not uncommon, is highly questionable from a health and welfare perspective. People who willingly feed cats they do not own are typically not as eager to see to their veterinary care [31]. Other concerns associated with feeding and interacting with wild and domesticated animals include the transmission of diseases and parasites. This includes transfer between animal populations, as well as transmission to humans. Common wildlife species, for example, hedgehogs, badgers, and wild birds are reservoirs for a wide range of zoonotic diseases, a risk that many of those who encounter them are unaware of [32,33]. There is also a significant risk of injury to humans and/or animals during feeding events, resulting from animals fighting to access food. During the COVID pandemic, these concerns were supplemented by the potential risk of COVID transmission between humans via animal fur [34].

Over half of respondents reported that their interactions with animals had not changed in regularity since the initial lockdown, and 5% reported that they had fewer interactions with animals since lockdown. This implies that lockdown did not cause a substantial increase in humans interacting with animals during their time outdoors, despite perceptions to the contrary. It may be that those who value animal encounters during their time outside always seek these interactions, while those who do not, did not tend to change this behaviour following lockdown.

The Natural England survey findings highlight the value of connecting with nature during lockdown: 85% of their sample agreed that ‘being in nature makes me very happy’, 74% reported taking more time to notice and engage in nature, and 41% reported that ‘nature and wildlife is more important than ever for my wellbeing’ [35]. This is reflected in the findings of the present study. While 46% of respondents felt that they had physically benefitted from walking outdoors during lockdown, 60% reported that their mental health had benefited. The fresh air and spending time in nature were both reported as reasons they enjoyed walking by over 80% of respondents. A smaller number of respondents reported interactions with wildlife (31%) or other animals (15%) as a reason for their enjoyment. The physical and mental health benefits of getting outside and engaging with nature are being increasingly recognized and promoted through initiatives such as ecotherapy [36,37] and green and blue social prescribing [38,39]. Reconnecting with the natural world in this way is only likely to gain momentum as we move on from the pandemic. The challenge will be to enjoy the benefits to mental and physical wellbeing that nature offers, without compromising the wellbeing of the animals that occupy this space or the wider environment.

## 5. Conclusions

The initial COVID-19 lockdown in the UK impacted engagement with the outdoors, resulting in an increase in the frequency that people went for walks outside and changes in the locations that these walks occurred. Interactions with nature (including wildlife, livestock, and horses) formed an important part of the walk experience for many, primarily through observations but also through feeding and physical interactions with wildlife and other animals (for a minority of respondents). The value of encounters with nature for mental, and to a lesser extent physical, health was evident; yet there is a risk that the benefits to human wellbeing could come at a cost to the welfare of the animals they encounter. The reconnection with nature reported during the COVID-19 pandemic is likely to continue into the future, and efforts must be made to minimise the potential negative consequences that this may have to ensure both animal welfare and the long-term sustainability of these activities.

## Figures and Tables

**Figure 1 animals-11-02281-f001:**
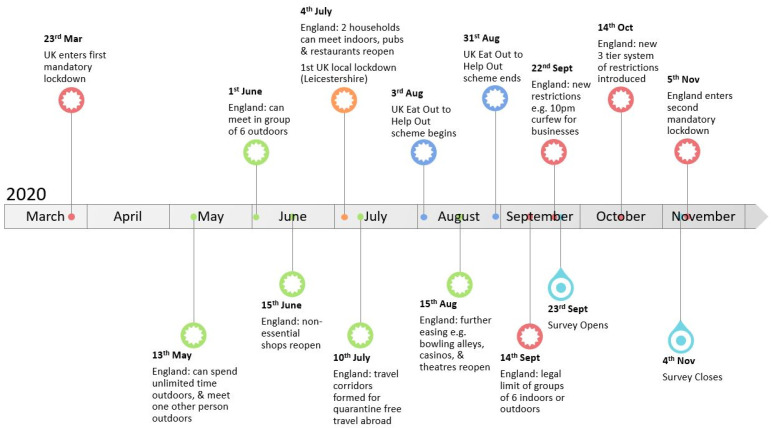
Summary of COVID-19 related restrictions and events in the UK, spanning the study period March-November 2020.

**Figure 2 animals-11-02281-f002:**
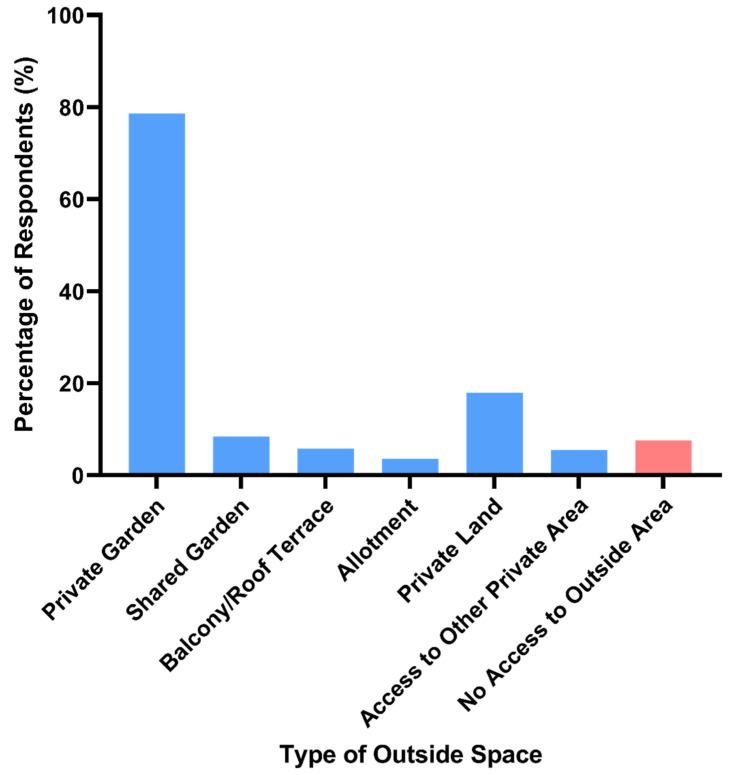
Percentage of respondents with access to different types of outside space, including those reporting to have no access to an outside area.

**Table 1 animals-11-02281-t001:** Frequency respondents reported going for walks before, during, and after lockdown.

Frequency	Pre-Lockdown n (%)	During Lockdown n (%)	Post-Lockdown n (%)
At least once a day	82 (26.7)	141 (45.9)	95 (30.9)
4–6 times/week	59 (19.2)	56 (18.2)	66 (21.5)
2–3 times/week	66 (21.5)	52 (16.9)	74 (24.1)
Once/week	43 (14.0)	25 (8.1)	39 (12.7)
Once every two weeks	25 (8.1)	7 (2.3)	10 (3.3)
Once/month	18 (5.9)	5 (1.6)	10 (3.3)
Less than once/month	12 (3.9)	6 (2.0)	8 (2.6)
Never	2 (0.7)	15 (4.9)	5 (1.6)

**Table 2 animals-11-02281-t002:** Respondent behaviour during walks in relation to location, who they walk with, and whether or not they stay on paths.

Survey Question and Related Answer Options		n	%
**Have you changed where you walk/spend time outside at all since the UK entered lockdown? (select all that apply)**			
	Yes, I have been walking closer to home	128	41.6
	Yes, I have been walking further from home	47	15.3
	Yes, I have been walking in more remote/rural areas e.g.,to avoid other people	93	30.2
	Yes, I have been walking in more urban areas	25	8.1
	Yes, I have been exploring a wider range of locations	94	30.5
	Yes, I have been sticking only to a few places I know well	35	11.4
	No change	71	23.1
**When you go out for walks, who do you usually go with? (Select all that apply)**			
	On your own	170	55.2
	With one other adult	198	64.3
	With a group of adults	37	12.0
	With children	46	14.9
	In a group with multiple adults and children	15	4.9
**When you are walking do you ever stray off the main path/footpaths/bridle paths? (Select all that apply)**			
	No	81	26.3
	Occasionally by accident	78	25.3
	Yes, if it provides a shortcut	43	14.0
	Yes, so I can explore/access areas I would otherwise be unable to reach	86	27.9
	Yes, as long as I am sure the land is not privately owned	75	24.4
	Yes, so I can interact with wildlife, horses or livestock	27	8.8
	Yes—other	15	4.9

**Table 3 animals-11-02281-t003:** Interactions with animals, including wildlife, horses and livestock, on walks as reported by respondents.

Survey Question and Related Answer Options		n	%
**When you go out for walks, do you ever interact with any wildlife in the following ways? (Wildlife refers to any wild animal that is not privately owned) (Select all that apply)**			
	I do not usually come across any wildlife	32	10.4
	I watch/look at wildlife	262	85.1
	I feed wildlife	31	10.1
	I physically interact with wildlife e.g., touching/picking up	14	4.5
	No, I do not interact with wildlife in any way	12	3.9
**When you go out for walks, do you ever interact with any other animals (e.g., livestock, horses) in the following ways? (This does not include dogs being walked by their owners, or animals owned by you) (Select all that apply)**			
	I do not usually come across any other animals	51	16.6
	I watch/look at other animals	206	66.9
	I feed horses	13	4.2
	I feed livestock	2	0.6
	I feed animals other than horses or livestock	5	1.6
	I physically interact with other animals e.g., touching/picking up	21	6.8
	No, I do not interact with other animals in any way	43	14.0
**At any point since the UK entered lockdown and up until now, have you been interacting with animals more regularly than you did before? (Select all that apply)**			
	I interact with animals less than I did before	16	5.2
	No change	173	56.2
	I watch animals more	93	30.2
	I physically interact with wildlife more	10	3.2
	I physically interact with horses more	9	2.9
	I physically interact with livestock more	4	1.3
	I feed wildlife more	11	3.6
	I feed horses more	2	0.6
	I feed livestock more	1	0.3

**Table 4 animals-11-02281-t004:** Respondent answers to questions relating to beneficial effects of walking and what they enjoy most about being outside.

Survey Question and Related Answer Options		n	%
**If you have been walking more since lockdown occurred, do you feel you have benefitted from this? (Select all that apply)**			
	No change	58	18.8
	Unsure	17	5.5
	Physical heath has benefitted	143	46.4
	Mental health has benefitted	186	60.4
	Negative impact on physical health	3	1.0
	Negative impact on mental health	3	1.0
	Other	2	0.6
**What do you most enjoy about walking/spending time in the countryside/green spaces? (Select all that apply)**			
	The exercise	235	76.3
	The fresh air	276	89.6
	Having a change of scenery	243	78.9
	Spending time in nature	248	80.5
	Interacting with wildlife	96	31.2
	Interacting with other animals	47	15.3
	Other	63	20.5

## Data Availability

The anonymised data are available on request from the corresponding author.

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
