# Peer review of "How Has the COVID-19 Pandemic Affected the Way We Access and Interact with the Countryside and the Animals within It?"

_animals, 2021, doi:10.3390/ani11082281_

Round 1

Reviewer 1 Report

Authors aims to quantify how lockdown affected human exercising behaviour, investigate what people did on their walks and to better understand what features of walks were important for the people.

Specific comments/suggestion are highlighted in the attached .pdf file.

Further information should be incorporated in terms of the Analysis subsection and a table could be added to the results summarizing the statistical analysis that is only mentioned in the text.

In the discussion, more information should be incorporated in terms of the risk associated to wild animals contact, there is not only a risk trough the feeding, but also on the potential zoonoses and reverse zoonoses events that could emerge from the contact with wild life.

Supplementary file is ok!

Author declare the limitation or skewed results on respondents living in England, this is very much appreciated.

The topic is important, to understand how this pandemic event has changed outdoor activities in humans and its potential contact with animals (wildlife, livestock and others).

Author Response

Comments and Suggestions for Authors

Authors aims to quantify how lockdown affected human exercising behaviour, investigate what people did on their walks and to better understand what features of walks were important for the people.

Specific comments/suggestion are highlighted in the attached .pdf file.

Thank you for bringing these to out attention, we have addressed your comments/suggestions in our revised manuscript.

Further information should be incorporated in terms of the Analysis subsection and a table could be added to the results summarizing the statistical analysis that is only mentioned in the text.

We have added further detail to the Analysis section so it now reads as follows: The quantitative data underwent descriptive analysis in SPSS using the frequencies function to ascertain the distribution of responses. As the survey was comprised of closed, categorical questions the data were not normally distributed; this was confirmed by visual inspection of histograms and Q-Q plots alongside the Kolmogorov-Smirnov test and Shapiro-Wilk test, both of which indicated that the data did not meet the criteria for normality (p≤0.001). Consequently, non-parametric tests were chosen for analysis. Changes in walking behaviour pre, during and post lockdown were analysed statistically using a Friedman Test, and post hoc Dunn tests with a Bonferroni correction were run between each pairing to explore where any differences in walking frequency were be-tween these three time-points lay. Chi-square tests for independence were used to investigate whether interaction with wildlife, livestock, horses and other animals was associated with walking alone, walking with other adults or walking with children.

We have chosen not to add a table summarising the results of the statistical analysis as only two research questions were explored statistically – the change in walking frequency across three time-points (pre-lockdown, during lockdown, post-lockdown) and whether who people walked with affected their interactions with animals on their walks. As there were relatively few statistical results to report it is clearer to leave them in the appropriate section of the text than to move them into tables.

In the discussion, more information should be incorporated in terms of the risk associated to wild animals contact, there is not only a risk trough the feeding, but also on the potential zoonoses and reverse zoonoses events that could emerge from the contact with wild life.

We briefly mentioned zoonoses in the original draft but have added the following sentence to expand on this: Common wildlife species, for example hedgehogs, badgers and wild birds, are reservoirs for a wide range of zoonotic diseases, a risk that many of those who encounter them are unaware of (Ruszkowski et al 2021; Simpson 2008).

Supplementary file is ok!

Thanks

Author declare the limitation or skewed results on respondents living in England, this is very much appreciated.

We have added an extra sentence explaining who this skew may have arisen: This skew in the sample is likely to stem from the recruitment strategy used which was biased towards English social media groups and mailing lists as these were more easily accessed by the research team who were all based in England

The topic is important, to understand how this pandemic event has changed outdoor activities in humans and its potential contact with animals (wildlife, livestock and others).

We agree – thank you for your constructive feedback on our paper.

Figure 1 – On the manuscript pdf, Reviewer 1 requested that the figure be created in SPSS as that is the platform used for our statistical analysis. However, due to quality issues we have previously encountered when trying to incorporate SPSS graphs into journal papers we made this figure in Excel. We hope that is acceptable to the editors.

Reviewer 2 Report

In my opinion, the present paper is interesting and is overall well written. I enjoyed reading it as concepts were concise and clear and stats were appropriate for the design that was developed.

Please mind the comments made below.

Simple summary. Well written. May put a little effort on highlighting the part of the study which is related to animals, give that as it is now, it seems to have been a very marginal part of it. If the journal if animals this is where the focus must be put.

Summary. Please, reduce by 129 words according to journal guidelines, summary length must be 200 words. I would like to see numbers and stats supporting the findings of authors.

Keywords. Try not to use the same words already used in your paper. Keywords are made to boost the potentiality of your article to be found in search engines. These engines already use the words in the title; thus, it makes no sense to repeat as you lose chances to be found.

Introduction. I enjoyed reading the introduction. I do not particularly see why making the distinction between livestock and horses if horses are livestock.

I personally think the objectives of a paper must be written on a single paragraph rather than enumerated.

I think this section may be complemented by adding a timeline graph on the progress of lockdown and ease of measurements in the UK, which is where the survey was developed.

Assumption testing needs to be provided. I agree that the most likely approach to be used here may be the most appropriate. Still, normality and homoscedasticity results must be added.

Line 140. Specify which routines in SPSS were used for each specific part.

Line 141. It is Friedman test not Freidman.

Line 142, it is Dunn test for pairwise comparisons and Bonferroni correction for significance.

Line 198. Could you provide a measure or graph for the skewness in the sample. This may help picture the specific nature of the sample used.

Discussion overall was really entertaining. I feel however, that results were a little too much descriptive sometimes.  Please add significance levels along discussion.

Author Response

In my opinion, the present paper is interesting and is overall well written. I enjoyed reading it as concepts were concise and clear and stats were appropriate for the design that was developed.

Thank you we appreciate this feedback.

Please mind the comments made below.

Simple summary. Well written. May put a little effort on highlighting the part of the study which is related to animals, give that as it is now, it seems to have been a very marginal part of it. If the journal if animals this is where the focus must be put.

The simple summary has been revised as follows: In March 2020, the UK entered its first mandatory lockdown to reduce the spread of COVID-19. The restrictions associated with the lockdown led to changes in human behaviour, particularly in relation to exercise and accessing the outdoors. This study used an online survey of UK resi-dents to explore these changes and to understand how people interacted with natural world while they were outside, including interactions with wildlife and domestic animals. There was an increase in how often respondents went for walks outside during lockdown compared to pre-lockdown levels and this increase lasted beyond the end of the first lockdown. Interacting with animals was a common feature of walks. This most commonly involved watching wildlife or domestic animals, such as livestock or horses, but sometimes also involved physically interacting with them and/or feeding them which may have implications for their welfare, for example if inappropriate food is provided. There is also a risk of humans transferring disease between the animals they have contact with, or zoonotic disease transmission between humans and animals. While is it positive that people are interacting with the natural world, it is important that this does not compromise animal wellbeing.

Summary. Please, reduce by 129 words according to journal guidelines, summary length must be 200 words. I would like to see numbers and stats supporting the findings of authors.

We have substantially revised the abstract and it now stands at 198 words: There is growing evidence that changes in human behaviour resulting from the COVID-19 pandemic have had positive and negative impacts on the natural world. This study used an online survey to explore how the first UK lockdown affected human exercising behaviour with particular focus on the role of wild and domestic animals. The survey was completed by 308 respondents. There was a significant increase in the frequency that respondents went for walks outdoors during lockdown in comparison to pre-lockdown levels (p≤0.001), and this was sustained albeit to a lesser extent, once lockdown ended (p=0.005). Engaging with the natural world was an important feature of walks outside for 81% of respondents. A small proportion of respondents reported physically interacting with the animals they encountered and/or feeding them which may have implications for their welfare. The findings suggest that those who value animal encounters during their time outside always seek these interactions, while those who do not did not tend to change this behaviour following lockdown. Should the changes in human exercising behaviour be sustained, it is important to balance the benefits of walking outdoors for human health and wellbeing, with the health and welfare of the animals they encounter.

Keywords. Try not to use the same words already used in your paper. Keywords are made to boost the potentiality of your article to be found in search engines. These engines already use the words in the title; thus, it makes no sense to repeat as you lose chances to be found.

We have now deleted COVID-19 from our keywords as that was the only keyword that was also in our title.

Introduction. I enjoyed reading the introduction. I do not particularly see why making the distinction between livestock and horses if horses are livestock.

In the UK, horses are not considered livestock which is why we made this distinction. We have added the following to the methods section (line 135-137) to clarify why we made the distinction: These three groups of animals were chosen based on anecdotal reports of humans engaging with these types of animals whilst on walks. In the UK horses are not considered a livestock species and so they were grouped separately in the survey.

I personally think the objectives of a paper must be written on a single paragraph rather than enumerated.

We have now restructured this into a single paragraph: This study contributes to the growing evidence base, using an online survey to explore human activity during the initial lockdown in the UK. The study had three main objectives; to quantify how lockdown affected human exercising behaviour (walking) and use of green spaces, to investigate what people did on their walks, including their interactions with wild and domestic animals (livestock and horses), and to better understand what features of walks were important for the people taking them during lockdown.

I think this section may be complemented by adding a timeline graph on the progress of lockdown and ease of measurements in the UK, which is where the survey was developed.

Thank you for this suggestion. We have now incorporated a timeline into the methods section of the paper.

Assumption testing needs to be provided. I agree that the most likely approach to be used here may be the most appropriate. Still, normality and homoscedasticity results must be added.

This has been done – we have added further detail to the Analysis section so it now reads as follows: The quantitative data underwent descriptive analysis in SPSS using the frequencies function to ascertain the distribution of responses. As the survey was comprised of closed, categorical questions the data were not normally distributed; this was confirmed by visual inspection of histograms and Q-Q plots alongside the Kolmogorov-Smirnov test and Shapiro-Wilk test, both of which indicated that the data did not meet the criteria for normality (p≤0.001). Consequently, non-parametric tests were chosen for analysis. Changes in walking behaviour pre, during and post lockdown were analysed statistically using a Friedman Test, and post hoc Dunn tests with a Bonferroni correction were run between each pairing to explore where any differences in walking frequency were be-tween these three time-points lay. Chi-square tests for independence were used to investigate whether interaction with wildlife, livestock, horses and other animals was associated with walking alone, walking with other adults or walking with children.

Line 140. Specify which routines in SPSS were used for each specific part.

This has been changed (see above).

Line 141. It is Friedman test not Freidman.

This has been changed (see above).

Line 142, it is Dunn test for pairwise comparisons and Bonferroni correction for significance.

This has been changed (see above).

Line 198. Could you provide a measure or graph for the skewness in the sample. This may help picture the specific nature of the sample used.

The percentage of respondents from each part of the UK are given at the start of the results section (Line 175-177 - 91.1% England, 3.0% Wales, 5.2% Scotland and 0.7% Northern Ireland). While we could add a pie chart to illustrate the spread of respondents across locations, I am not sure it would really add anything when there are only four countries within the UK and the percentages given demonstrate how unbalanced the sample was.

Discussion overall was really entertaining. I feel however, that results were a little too much descriptive sometimes.  Please add significance levels along discussion.

We are glad you liked the discussion. We have made sure that percentage values are reported in the discussion to illustrate the points we are making in the text. However, reporting significance levels in the discussion is not normal practice, usually is it enough in the discussion to state that significant differences or associations were found or not found. In our paper, only two research questions were explored statistically and the significant results are stated as such in the discussion.

Round 2

Reviewer 1 Report

Author's have made all the modifications and suggestions indicated by the reviewers, improving the quality of the MS.

Specific comments are highlighted in the attached .pdf file.

One specific query keeps in mi mind in terms of this sentence "Over half of respondents (55.2%, n=170) walked alone while 64.3% (n=198) walked with another adult."

Author Response

We have revised Figure 2 and re-phrased the sentence on who respondents walked with in the results sections (Line171-173) as requested. Please see manuscript with tracked changes attached here.
